# Bayesian variable selection for genome-wide association study of grain traits in rice

Rupam Basu[1], Sabyasachi Mukhopadhyay[2], Kaustubh Adhikari[3]*

1 Decision Sciences, Indian Institute of Management Udaipur, Udaipur, Rajasthan, India, 2 Operations Management, Indian Institute of Management Calcutta, Kolkata, West Bengal, India, 3 School of Mathematics and Statistics, Open University, Milton Keynes, United Kingdom

☯ These authors contributed equally to this work.
* kaustubh.adhikari@open.ac.uk

## Abstract

Rice (Oryza sativa) is a staple food crop for more than half of the world's population. Besides high gluten-free nutritional contents, it has high economic value supporting livelihood of millions of farmers. That is why a lot of research is being carried out to derive new varieties of rice and improve its yield, stress tolerance, and grain quality. It remains a central goal in agricultural research. Genome-wide association studies (GWAS) provide a powerful framework for linking genetic variation to complex phenotypic traits, but the high dimensionality of genomic data presents significant challenges for model selection and prediction. Using rice genotype and phenotype data, we compared the performance of several frequentist and Bayesian modeling approaches: multiple linear regression (OLS: Ordinary Least Squares), LASSO (Least Absolute Shrinkage and Selection Operator), Ridge, Bayesian LASSO, Bayesian Sparse Linear Mixed Model (BSLMM), and a Bayesian spike-and-slab prior model. Phenotypic traits were transformed where necessary to approximate normality, and predictive performance was evaluated through cross-validation using mean squared error and predictive correlation. The spike-and-slab prior model often outperformed the classical methods, yielding superior prediction and effective variable selection. Our findings demonstrate the value of Bayesian model selection frameworks for plant GWAS and trait prediction, and highlight the effectiveness of Bayesian methods in identifying informative markers in rice. Such approaches hold promise for accelerating genetic improvement and supporting marker-assisted selection in crop breeding programs. Rather than emphasizing biological interpretation of individual loci, our results highlight differences in predictive behavior, stability, and inferential characteristics across models.

**Data availability statement:** The data is publicly availabe from Orhobor et al. 2018, as mentioned in the manuscript. The URL is https://data.mendeley.com/datasets/sr8zzsrpcs/1.

**Funding:** The author(s) received no specific funding for this work.

**Competing interests:** The authors have declared that no competing interests exist.

# 1 Introduction

Rice (Oryza sativa) is a staple food crop for more than half the world's population, playing a central role in global food security. In addition to its nutritional values, economic importance of the crop is also very high for supporting livelihoods of millions of farmers. Improving rice yield and quality traits through breeding has been a major goal of agricultural genetics. Genome-Wide Association Studies (GWAS) have emerged as a powerful strategy for identifying genetic variants, such as Single Nucleotide Polymorphisms (SNPs), that are associated with agronomic traits of interest [1,2].

The genomic revolution has generated massive amounts of genetic data that have facilitated research regarding genotype and phenotype relationship, that is, genetic set-up about observable characteristics. With the advent of genetic pathway data, new statistical methods have received considerable advancement in light of the growing supply and availability of high-dimensional genomic data. Historically speaking, traditional genome-wide association studies (GWAS) have formed the pillar of identifying genetic variants correlated with traits. Genome-wide association studies (GWAS) have historically been used to establish the association between individual SNPs and traits of interest in this context [3]. discusses the impact of GWAS over five years, highlighting its successes in discovering common genetic variants associated with complex traits [4], presents efficient imputation techniques that leverage linkage disequilibrium between SNPs to improve the accuracy of phenotype prediction. The research by [5] highlights the power of GWAS in identifying common variants associated with disease phenotypes and demonstrates the effectiveness of association tests in large-scale data.

GWAS is based on the single SNP model, in which each SNP is independently tested for association. However, this approach has several drawbacks: multiple testing burdens, lack of power to detect small effect sizes, and failure to account for interactions and polygenic architectures [3,6]. Lastly, most GWAS approaches assume that all SNPs contribute additively to the trait of interest rather than to more complex genetic relationships [7].

To address these issues, both frequentist and Bayesian methods have been developed. Frequentist approaches, such as the LASSO (Least Absolute Shrinkage and Selection Operator) [8] and Elastic Net [9], have been widely adopted for their ability to handle high-dimensional data, providing variable selection and shrinkage to improve predictive performance. These methods extend linear models by adding penalty terms to the regression coefficients. LASSO [8] imposes a $L_1$-penalty, leading to sparsity, where many coefficients are shrunk to zero, making it useful for high-dimensional data. Elastic Net combines $L_1$-penalty (LASSO) and $L_2$-penalty (Ridge regression) [10], helping to handle correlated features (SNP).

Linear Mixed Models (LMM) [11] are frequently used in genetics to account for population structure and relatedness among samples, providing improved control over false positives. It allows for the inclusion of random effects that account for population structure and relatedness between individuals. These models are used mainly in genetics to ameliorate GWAS by confounding genetic relatedness. However,

subsequently, LMMs typically assume heterogeneous effects across SNPs, which often fails to capture the complexity of the data.

Another limitation of frequentist methods (such as LASSO and LMMs) in modifying predictions is that they lack the flexibility to assume prior biological information and model a complex genetic structure. Because Bayesian methods are very flexible in modeling complex data structures while incorporating prior knowledge and dealing with uncertainty, their use in genetics is spreading rapidly.

Bayesian methods, on the other hand, offer a flexible framework for incorporating prior knowledge and robustness in handling the complexity of high-dimensional data. Bayesian variable selection techniques, such as the spike and slab model [12] and Bayesian LASSO [13], have proven effective in genetic studies, particularly for addressing the sparsity and high-dimensionality typical of such datasets. Furthermore, Bayesian linear mixed models (Bayesian LMMs) [14,15] and hybrid models like the Bayesian Sparse Linear Mixed Model (BSLMM) [16] combine fixed and random effects to improve predictive accuracy and SNP selection. These models enable the probabilistic selection of the relevant SNPs and provide uncertainty quantification of the effects of these SNPs on phenotypic traits.

In this study, we will compare six models for phenotype prediction: the multiple linear regression model as a baseline, ordinary LASSO, ordinary ridge regression, the spike and slab regression model, the Bayesian LASSO (BLASSO), and the Bayesian Sparse Linear Mixed Model (BSLMM). By using these models to predict grain length, grain width, and grain weight from the Rice genotype data, we aim to evaluate their relative performance concerning their ability to handle high-dimensional genomic data, identify relevant SNPs, and yield accurate phenotype predictions.

Although the data considered arise from genome-wide association studies, this work is not intended as a comprehensive GWAS aimed at biological discovery. Instead, the emphasis is on methodological comparison and predictive performance of high-dimensional regression models, with biological interpretation of specific variants considered beyond the scope of the present study.

The remainder of the paper is arranged in the following way. In Section 3, we have described the data and its components. In Section 3 we have discussed the various Bayesian models under considerations. A brief description of the MCMC techniques used for simulating from the Bayesian models is given in Section 3.5. In Section 5, we have described the cross-validation method used for comparing the models and the results of the cross-validation exercise using the data. Finally, we have discussed the overview of our findings and the scope of future research in Section 6.

## 2 Data

We use the rice genotype and phenotype data set compiled and described by [17], which comprises measurements from 2,266 rice plants drawn from diverse accessions. The dataset includes 12 phenotypic traits relevant to agronomic performance and grain characteristics, as well as 12,486 single-nucleotide polymorphism (SNP) markers obtained through genotyping-by-sequencing (GBS) technology, with three important agronomic traits in rice breeding programs: grain length, width, and seedling height.

In our analysis, we will use the three phenotypes: **GRLT (Grain length)**, **GRWD (Grain width)**, and **SDHT (Seedling height)** as response variables. Genotypic markers will serve as covariates to explain phenotypes. The summary of the three variables is shown in Table S1 in S1 Text of the Supplementary Materials. The biplot of the three phenotypes—grain length (GRLT), grain width (GRWD), and seedling height (SDHT)—shows that the first two principal components explain 37.7% and 33.5% of the total variation, respectively (Fig 1). Grain width contributes primarily to the second principal component, whereas grain length and seedling height are more closely aligned with the first principal component, indicating distinct dimensions of phenotypic variation. On the other hand, the biplot of SNPs showed that the first two principal components explained 29.1% and 4.1% of the total genetic variation (Fig 2). The spread of points illustrates the genetic structure of the population, with distinct clustering patterns suggesting underlying variation among samples.

Since normality of the regressed variables is an important feature that facilitates the Genome-Wide Association Study (GWAS), we performed normality tests for three phenotypes: **GRLT (Grain length)**, **GRWD (Grain width)**, and **SDHT**

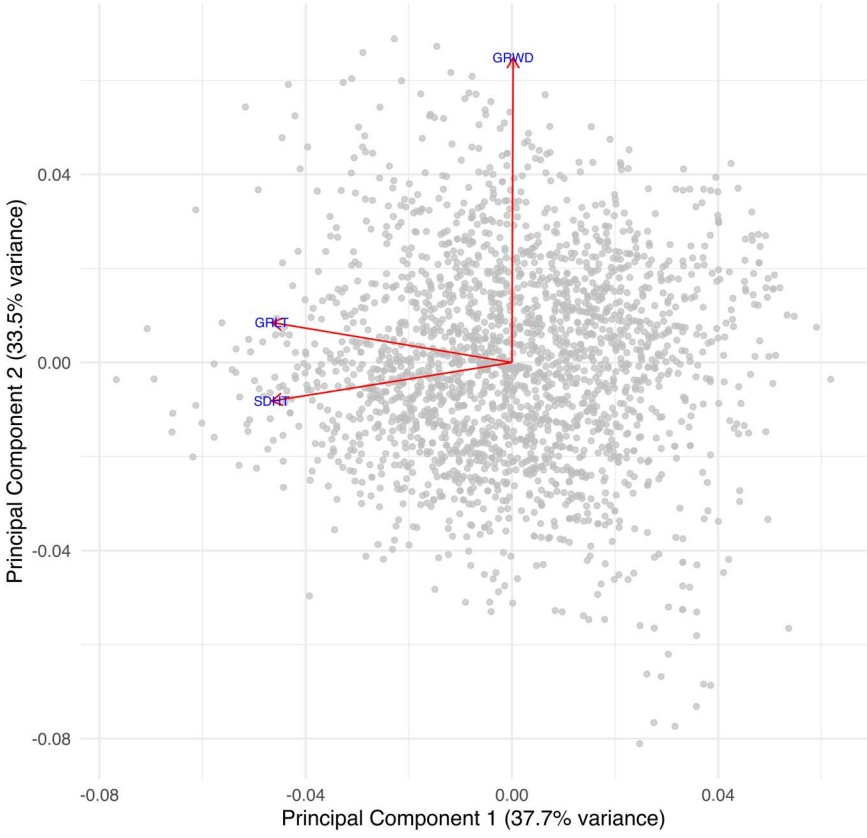

**Fig 1. Biplot of the phenotypes.** Biplot of grain length (GRLT), grain width (GRWD), and seedling height (SDHT) showing the distribution of samples along the first two principal components (37.7% and 33.5% of total variation, respectively). Vectors indicate the contribution and orientation of each phenotype in the reduced component space.

**(Seedling height)** (see Section S1.1 in the Supplementary Materials in S1 Text). While **GRWD (Grain width)** showed the presence of normality in the data, **GRLT (Grain length)** and **SDHT (Seedling height)** indicate a departure from normality. For the latter two variables, we applied *Order Quantile Normalization (OQN)*. In Section S1.1 in S1 Text we have shown how this transformation achieved normality for the two remaining variables.

## 3 Methodology: Bayesian modeling

We begin by considering a simple linear model that relates phenotypes $y$ to genotypes $X$. Specifically, $y = (y_1, y_2, \ldots, y_n)^T$ is an $n$-dimensional vector of phenotypes measured on $n$ entities, and $X = (x_1, x_2, \ldots, x_n)^T$ is an $n \times p$ matrix of genotypes corresponding to these same entities at $p$ (=12,486 in our data) genetic markers. The vector $\beta \in \mathbb{R}^p$ contains the (unknown) effects of the genetic markers.

In this work, we have considered four hierarchical structures of the four prominent Bayesian models in GWAS: the spike-and-slab regression model, the Bayesian LASSO, the Bayesian linear mixed model, and the Bayesian Sparse Linear Mixed Model.

### 3.1 Spike and slab regression model

In the first model, we used spike-and-slab priors [12] on the regression coefficients $\beta \in \mathbb{R}^p$, which is a Bayesian variable-selection technique that allows sparsity to be introduced by assigning a probability of being included in the model to each

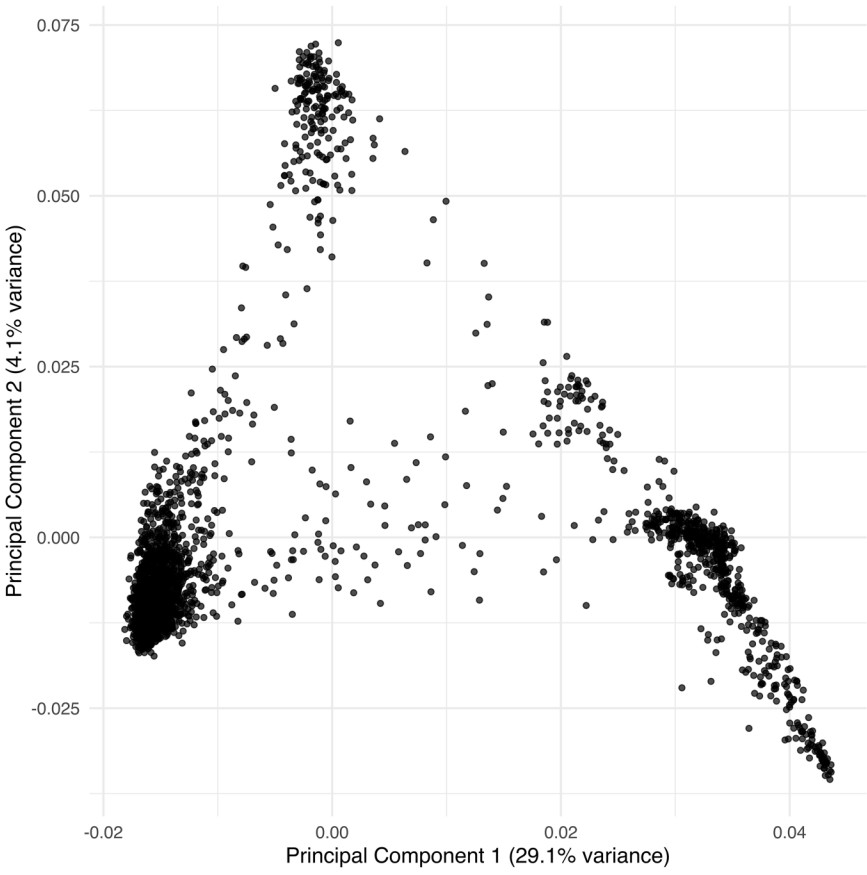

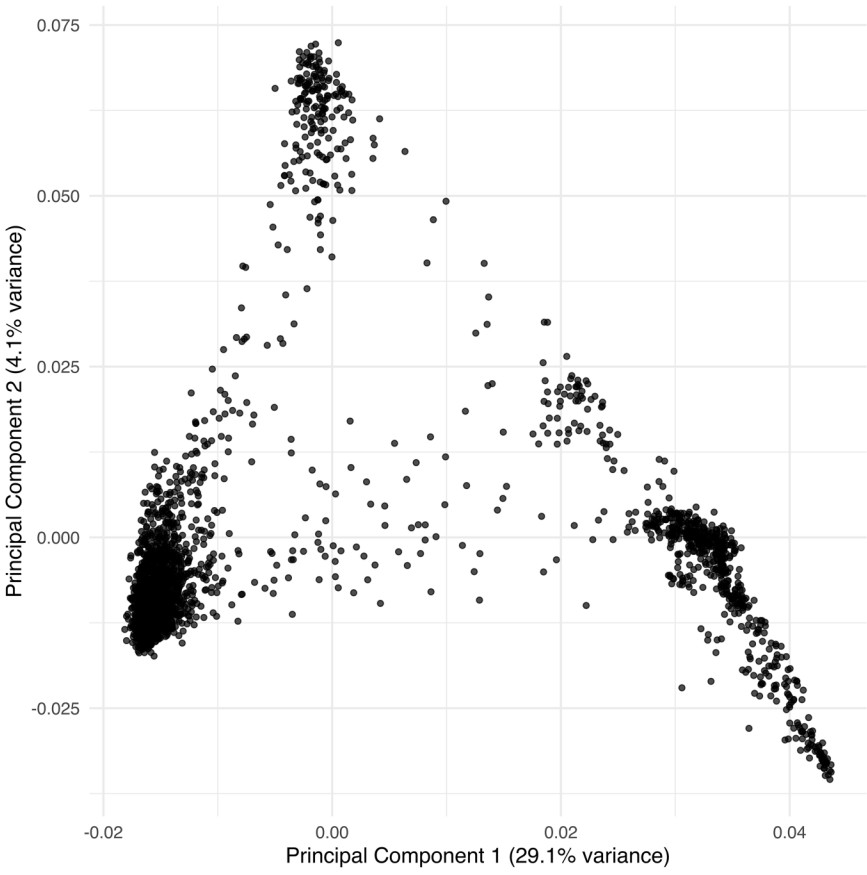

**Fig 2. Principal components plot of the SNPs.** PCA plot of single nucleotide polymorphisms (SNPs) showing the distribution of samples along the first two principal components, which explain 29.1% and 4.1% of the total variance, respectively.

SNP. It is highly suitable for high-dimensional genetic data in which only a few SNPs may contribute to the phenotype. The spike-and-slab prior is, in fact, a mixture model that has both a point mass at a point very close or precisely to zero (the spike) and a diffuse distribution (the slab), producing variable selection with regularization [18]. The spike component encourages sparsity by shrinking irrelevant coefficients to zero, effectively performing variable selection [12]. The slab component allows for including relevant predictors with proper regularization, preventing overfitting [12]. Importantly, Bayesian methods inherently account for model uncertainty, providing probabilistic statements about including predictors [18].

The hierarchical structure is as follows:

$$y_i = \mathbf{x}_i^{\mathsf{T}}\beta + \epsilon_i, \quad \epsilon_i \sim N(0, \sigma_\epsilon^2),$$
$$\left(\beta_j \mid \gamma_j, \sigma^2\right) \overset{\text{ind}}{\sim} N\left(0, \gamma_j v_1 \sigma^2 + (1 - \gamma_j) v_0 \sigma^2\right), \quad j = 1, \ldots, k,$$
$$\left(\gamma_j \mid \pi\right) \overset{\text{i.i.d.}}{\sim} \text{Bernoulli}\left(\frac{1}{2}\right),$$
$$\left(\sigma^2 \mid \gamma, \nu, \lambda\right) \sim \text{IG}\left(\frac{\nu}{2}, \frac{\nu\lambda}{2}\right),$$
$$\left(\sigma_\epsilon^2 \mid a, b\right) \sim \text{IG}\left(a, b\right)$$

(1)

In the above model:

- $\beta_j$ represents the regression coefficients.

- $\gamma_j$ is an indicator variable that determines whether $\beta_i$ is included in the model (spike-and-slab).

- $v_0$ and $v_1$ are variance parameters for the spike (small value) and slab (larger value) components, respectively.

- $\sigma^2$ is the variance parameter for the distribution of the slab part of $\beta_j$, following an inverse gamma distribution with shape parameter $\frac{\nu}{2}$ and scale parameter $\frac{\nu\lambda}{2}$.

- $\pi$ is the inclusion probability of $\gamma_i$.

- $\sigma^2_\epsilon$ is the variance of the error terms.

- IG stands for Inverse-Gamma distribution.

The spike-and-slab prior mentioned above is a continuous bimodal prior, with $v_0$ = 0.00025 representing a small near-zero value and $v_1$ = 1. The hyperparameters $\nu$ = 10 and $\lambda$ = 125 are the shape and scale parameters for the inverse gamma distribution for $\sigma^2$, and the parameters $a = b = 0.001$ represent the shape and scale parameters for the inverse gamma distribution for $\sigma^2_\epsilon$. These parameters are chosen so that $\gamma_i\tau_i^2$ has a continuous bimodal distribution with a spike at $v_0$ and a right continuous tail. The parameter $\pi$ represents the inclusion probability [12].

### 3.2  Bayesian LASSO

The Bayesian LASSO [13] imposes a Laplace (double-exponential) prior on the regression coefficients, inducing shrinkage of the SNP effects. This model is suitable for situations where many minor effects are spread across the genome.

The hierarchical structure is as follows:

$$
\begin{aligned}
y_i &= \mathbf{x_i^T}\beta + \epsilon_i, \quad \epsilon_i \sim \mathrm{N}(0,\sigma^2) \\
\beta_j &\sim \mathrm{Laplace}(0,\lambda), \quad j = 1,\ldots,p \\
\lambda &\sim \mathrm{Gamma}(a_\lambda, b_\lambda) \\
\sigma^2 &\sim \mathrm{Inverse\ Gamma}(a_\sigma, b_\sigma)
\end{aligned}
\tag{2}
$$

The hyperparameters $a_\lambda$ = 0.55 and $b_\lambda$ = $10^{-6}$ are the shape and scale parameters for the gamma distribution for $\lambda$, and the parameters $a_\sigma = b_\sigma$ = 0.5 represent the shape and scale parameters for the inverse gamma distribution for $\sigma^2$. In this case, $\lambda$ is the regularization parameter controlling the amount of shrinkage. The Bayesian LASSO is helpful in genetic studies because it can shrink the effect sizes of irrelevant SNPs, making it suitable for polygenic traits with many minor effects.

### 3.3  Bayesian linear mixed model (BLMM)

The Bayesian linear mixed model [19] extends the standard linear model by incorporating random effects to account for population structure and genetic relatedness, which are often critical in genetic studies.

The hierarchical structure is as follows:

$$
\begin{aligned}
y_i &= \mathbf{x_i^T}\beta + \mathbf{Z_i}\mathbf{u} + \epsilon_i, \quad \epsilon_i \sim \mathrm{N}(0,\sigma^2) \\
\mathbf{u} &\sim \mathrm{N}(0, \mathbf{G}\sigma^2_u) \\
\beta_j &\sim \mathrm{N}(0, \tau^2), \quad j = 1,\ldots,p \\
\sigma^2 &\sim \mathrm{Inverse\text{-}Gamma}(a_\sigma, b_\sigma)
\end{aligned}
\tag{3}
$$

Here, $\mathbf{Z}_i\mathbf{u}$ represents random effects that capture population structure, and $\mathbf{G}$ is the genetic relationship matrix. The BLMM is advantageous for modeling fixed and random SNP effects, making it ideal for complex population structures.

### 3.4 Bayesian sparse linear mixed model (BSLMM)

The Bayesian Sparse Linear Mixed Model (BSLMM) [16] combines the features of both the linear mixed model (LMM) and sparse regression models, such as the spike and slab. The Bayesian Sparse Linear Mixed Model is beneficial for genetic studies because it models minor polygenic effects (as random effects) and significant individual SNP effects (as fixed effects). Such a combined model can accommodate the intricate genetic architecture of traits by balancing the inclusion of many minor effects with the selection of a few significant impacts during GWAS.

The hierarchical structure is as follows:

$$y_i = \mu + \mathbf{x}_i^T\beta + \mathbf{Z}_i\mathbf{u} + \epsilon_i,$$
$$\epsilon_i \sim N(0, \sigma_\epsilon^2 I_n),$$
$$\mathbf{u} \sim N(0, \sigma_b^2 K), K = \frac{1}{n}XX^T$$
$$\beta_i \sim \pi N(0, \sigma_a^2) + (1 - \pi)\delta_0,$$
$$\mu \sim N(0, \sigma_\mu^2),$$
$$\sigma_\epsilon^{-2} \sim \text{Gamma}(\alpha_\epsilon, \gamma_\epsilon),$$
$$\sigma_a^2 \sim \text{Inverse-Gamma}(\alpha_a, \gamma_a),$$
$$\sigma_b^2 \sim \text{Inverse-Gamma}(\alpha_b, \gamma_b),$$
$$\log(\pi) \sim \text{Uniform}\left(\log\left(\frac{1}{p}\right), \log(1)\right)$$

(4)

The hyperparameters $\alpha_a = 10$ and $\gamma_a = 0.01$ are the shape and scale parameters for the inverse-gamma distribution for $\sigma_a^2$, $\alpha_b = 10$ and $\gamma_b = 0.01$ are the shape and scale parameters for the inverse-gamma distribution for $\sigma_b^2$ and the hyperparameters $\alpha_\epsilon = \gamma_\epsilon = 0.001$ represent the shape and scale parameters for the inverse-gamma distribution for $\sigma_\epsilon^2$. The hyperparameter $\sigma_\mu^2$ is taken as the variance of the phenotype measure under study.

The advantage of BSLMM is that it provides a flexible framework that can handle both sparse and polygenic architectures. It combines the strengths of the LMM in capturing polygenic effects with the spike and slab model for SNP selection. This is particularly relevant in genetic studies where both large and small effects play a role in determining the phenotype. y adjusting the proportion of SNPs that have non-zero effects ($\pi$) and the magnitude of random effects ($\sigma_u^2$), BSLMM can adapt to a variety of genetic architectures. Including the matrix $K$ allows the model to account for population structure and relatedness among individuals, reducing confounding in genetic association studies.

The BSLMM is a powerful tool for both phenotype prediction and estimating the proportion of variance explained (PVE) by genotypes, which is an improvement over traditional LMMs and sparse regression models.

In the following sections, we will discuss the analysis performed to compare the prediction performance of several models, including the Spike and Slab regression model, Bayesian LASSO (BLASSO), and Bayesian Sparse Linear Mixed Model (BSLMM), against the traditional multiple linear regression model. We will begin with a brief overview of the dataset, followed by a description of the models and the methods used for posterior sampling. Finally, we will present and discuss the results of the comparison.

### 3.5 MCMC technique for posterior sampling

For the Bayesian variable selection models considered—specifically Spike and Slab, Bayesian LASSO, and BSLMM—inference relies on the marginal posterior distributions of the SNP effects, $\beta$, and the associated variance components.

However, due to the high dimensionality of the genotype data ($p \gg n$) and the non-conjugate nature of the sparsity-inducing priors (e.g., Laplace or point-mass mixtures), the joint posterior distribution $P(\beta|\mathbf{X}, \mathbf{y})$ is analytically intractable. The high-dimensional integrals required for normalizing the posterior density preclude direct evaluation.

Posterior sampling techniques are required because directly computing these posterior distributions involves solving integrals that are too complex to evaluate analytically. In such cases, Markov Chain Monte Carlo (MCMC) methods are used to generate samples from the posterior distribution, allowing us to approximate it through these samples.

In Bayesian models, we are interested in the posterior distribution of the model parameters, such as:

$$P(\beta|X, y) \propto P(y|X, \beta)P(\beta)$$

Where, $P(y|X, \beta)$ is the likelihood of the data given the parameters and $P(\beta)$ is the prior distribution of the parameters.

For models like the Spike and Slab, Bayesian LASSO, and BSLMM, the posterior distribution does not have a closed-form solution due to the complex interplay between priors and likelihoods, especially when sparsity-inducing priors like the Laplace prior or spike-and-slab prior are used.Consequently, we approximate the posterior distribution using MCMC methods. Specifically, we implemented a component-wise Gibbs sampling algorithm to generate samples from the full conditional distributions of the parameters. The derivation of these conditional distributions and the details of the implementation in the `R` statistical computing environment are provided in Section S2 of the Supplementary Materials in in S2 Text.

### 3.6 MCMC diagnostics

Posterior inference for the Spike and Slab regression model was conducted via MCMC sampling. To ensure the reliability of the approximation, we rigorously assessed the convergence and mixing properties of the chains using a combination of visual and quantitative diagnostics, adhering to standard protocols in Bayesian computation. We executed three independent MCMC chains with over-dispersed initial values and distinct random seeds. Each chain ran for 8,000 iterations, with the initial 3,000 iterations discarded as burn-in. All subsequent diagnostics were computed based on the post-burn-in samples. A visual inspection was performed using trace plots of key parameters, including the regression coefficients. Additionally, we examined posterior density plots to verify consistency across chains. Quantitative convergence was evaluated using the Gelman–Rubin potential scale reduction factor ($\hat{R}$) and Effective Sample Size (ESS) [20–22]. While diagnostics were computed for all model parameters, we focused our reporting on the regression coefficients, as these are the primary quantities of interest for variable selection.

Upon checking the convergence results and confirming that the chains have converged after the initial 3,000 iterations as burn-in, for the final models, the main MCMC chains were simulated for 25,000 iterations, with the first 5,000 iterations discarded as burn-in to mitigate the influence of initial values and ensure stationarity. MCMC inferences were based on the remaining 20,000 samples.

### 3.7 Prediction of phenotypes

To perform the cross-validation exercise as described in Section 3.5, we need to simulate from the posterior distributions of the three individual models. Subsequently, for each model, we need to predict the genotypes of the test data. The model whose predictions have the smallest deviation from the observed genotypes in the test data (with deviation measured by the metrics described in Section 4.1) is taken as the optimal model for these data.

For producing predictions, we used the idea of **Bayesian Model Averaging (BMA)** [23] for three competing models (denoted as $M_1$, $M_2$, $M_3$ and $M_4$). For a new phenotype $x^*$ in the test set, the predictive distribution under model $M_k$, $k = 1, 2, 3, 4$ is given by

$$p(y^* \mid x^*, D, M_k) = \int p(y^* \mid x^*, \beta, D, M_k) \times p(\beta \mid D, M_k)d\beta,$$

where $p(\beta \mid D, M_k)$ denotes the posterior distribution of parameters $\beta$ given training data $D$ for model $M_k$.

If $\beta^{(1)}, \ldots, \beta^{(L)}$ be the $L$ MCMC simulations after burn-in, then using Monte Carlo method we can get estimate of the predictive distribution as

$$\hat{p}(y^* \mid x^*, D, M_k) = \frac{1}{L} \sum_{\ell=1}^{L} p(y^* \mid x^*, \beta^{(\ell)}, D, M_k)$$

for $k$-th model.

For each of the Bayesian methods, such as Spike and Slab, BSLMM, and Bayesian Lasso, we simulate $\beta^{(\ell)}$ from the posterior distributions and obtain the response predictions at a new point $x^*$ by simulating $y^{*(1)}, \ldots, y^{*(L)}$ from $p(y^* \mid \beta^{(\ell)}, x^*, M_k)$ and averaging over all simulations. So, the predicted value of $y^*$ given $x^*$ is given by

$$\widehat{y}^* = \frac{1}{L} \sum_{\ell=1}^{L} y^{*(\ell)}$$

## 4 Methodology: comparison between models

We follow the conventional approach of splitting the data, where 80% (1,814 observations) is used to train the model, that is, to derive the posterior distribution, and the remaining 20% (454 observations) is reserved as test data for predicting phenotypes and evaluating model performance.

For each of the three models 1, 2 and 4 we used training data to obtain posterior mean estimates of SNP effects to computed predicted phenotypes for individuals in the test set. We compared the predicted and observed values of the phenotypes in the test set using various metrics (see Section 4.1) for checking the performance of the individual models.

### 4.1 Metrics for model performance

In evaluating the performance of the three Bayesian approaches (Spike-and-Slab, Bayesian LASSO, and BSLMM) in genetics, we consider several key metrics to assess their predictive accuracy and uncertainty:

- **Root Mean Squared Error (RMSE):** RMSE quantifies the square root of the average squared differences between predicted values $\hat{y}_i^*$ and actual observations $y_i^*$:

$$\text{RMSE} = \sqrt{\frac{1}{n} \sum_{i=1}^{n} (y_i^* - \widehat{y}_i^*)^2}$$

where $y_i^*$ denotes actual values for SNP $x_i^*$, $\widehat{y}_i^*$ signifies predicted values, and $n$ represents the number of observations in the test set. Lower RMSE values indicate better model fit.

- **Mean Absolute Error (MAE):** MAE computes the average magnitude of the errors between predicted and actual values:

$$\text{MAE} = \frac{1}{n} \sum_{i=1}^{n} |y_i^* - \hat{y}_i^*|$$

MAE provides a straightforward interpretation, reflecting the average error in the same units as the data.

- **Predictive Coverage:** Predictive coverage assesses the uncertainty of predictions, particularly relevant for Bayesian models. It involves constructing a predictive interval using the 5-th and 95-th quantiles of simulated values $y^{*(1)}, \ldots, y^{*(L)}$:

$$\text{Predictive Interval} = [\hat{y}^*(0.05), \hat{y}^*(0.95)]$$

Predictive coverage calculates the percentage of actual values $y^*$ falling within this interval, indicating the model's ability to capture uncertainty.

These metrics collectively provide a comprehensive evaluation of model performance, emphasizing accuracy and uncertainty estimation, which are critical for robust genetic analyses.

### 4.2 Cross-validation and comparative assessment

To rigorously assess the predictive performance of the Spike and Slab regression model against alternative Bayesian specifications, as well as frequentist Ridge and LASSO regression benchmarks, we employed a *k*-fold cross-validation scheme with *k* = 5. The dataset was randomly partitioned into five disjoint subsets of approximately equal size. In each iteration, four subsets were utilized for model training (representing 80% of the data), while the remaining subset (20%) served as the validation set for out-of-sample prediction. All statistical computations were implemented in the `R` statistical computing environment. For the Bayesian frameworks, we assessed performance based on RMSE, MAE, and the coverage probability of the 95% posterior predictive intervals. For the frequentist penalized regression models (Ridge and LASSO), performance was evaluated using RMSE and MAE. To provide a comprehensive view of model stability and predictive power, we report the mean, minimum, and maximum values for these metrics across the five folds. The detailed results of this comparative analysis are presented in Section 5.

### 4.3 Assessing model size

As discussed earlier, many SNP effect sizes are expected to be zero in three Bayesian models – 1, 2, and 4 due to their role in variable selection. Thus, we also record the model size, defined as the number of non-zero effect sizes (Number of variables), for each model.

The model size $S$ is defined as:

$$S = \sum_{j=1}^{p} \mathbb{I}(\beta_j \neq 0)$$

where $\mathbb{I}(\cdot)$ is the indicator function and $\beta_j$ represents the effect size of SNP $j$.

The model size can also be calculated for LASSO, as the number of non-zero effect sizes. It is not relevant for the other frequentist methods, OLS and Ridge, as these methods do not shrink any effect sizes to zero.

## 5 Results

### 5.1 MCMC diagnostics

We assessed the convergence and mixing properties of the chains using a combination of visual and quantitative diagnostics, adhering to standard protocols in Bayesian computation. Visual inspection of each of the three chains was performed using trace plots of key parameters, including the regression coefficients. These plots exhibited stable trajectories, good mixing, and no discernible trends or drifts, indicating satisfactory exploration of the posterior distribution. Additionally, we

examined posterior density plots to verify consistency across chains. Representative plots for the phenotypes **GRLT**, **GRWD**, and **SDHT** are provided in Section S3 of the Supplementary Material in S3 Text.

Quantitative convergence was evaluated using the Gelman–Rubin potential scale reduction factor ($\hat{R}$) and Effective Sample Size (ESS) [20–22]. Table 1 summarizes the minimum, maximum, and average $\hat{R}$ and ESS values calculated across the vector of regression coefficients for each of the three phenotypes. For all phenotypes, $\hat{R}$ values remained strictly below 1.01, satisfying the convergence threshold recommended by [20,21]. Regarding sampling efficiency, ESS estimates generally exceeded 1000. The only exception was the GRLT phenotype, where the minimum ESS was 900; however, this still exceeds the required bounds for reliable inference [21,22]. Collectively, these diagnostics provide strong evidence that the MCMC chains converged to the target stationary distribution.

## 5.2 Results of cross-validation exercise

For each of the three phenotypes, as mentioned earlier, we applied three classical (frequentist) models: OLS, LASSO, and Ridge regression, and three Bayesian models: Spike-and-Slab, Bayesian LASSO, and Bayesian Sparse Linear Mixed Model (BSLMM).

We compared the performance of the three competing Bayesian models using a five fold cross-validation method described in Section 4. In addition, we also compared the predictive accuracy of these models with classical frequentist methods for multiple linear regression.

First, we present the model size, defined as the number of non-zero effect sizes (Number of variables), for each model, in Table 2.

We also evaluate the models using metrics such as Root Mean Squared Error (RMSE), Mean Absolute Error (MAE), and predictive coverage, for models with different prior choices. It is important to note that predictive coverage does not apply to the three frequentist models, as it does not involve any prior assumptions about the coefficients. The results for these performance metrics are presented in Table 3 for **GRWD (Grain Width)**, Table 4 for **GRLT (Grain Length)**, and Table 5 for **SDHT (Seedling Height)**.

## 5.3 Conclusions from the cross-validation exercise

The cross-validation results for Grain Length (GRLT), Grain Width (GRWD), and Seedling Height (SDHT) are summarized in Tables 3, 4, and 5, respectively. These tables detail the performance of the Spike-and-Slab, BLASSO, and BSLMM models with respect to Root Mean Square Error (RMSE), Mean Absolute Error (MAE), and predictive coverage.

**Table 1. Summary of MCMC diagnostics. Gelman–Rubin ($\hat{R}$) and Effective Sample Size (ESS) statistics for the regression coefficients of each phenotype.**

| Phenotypes | Gelman–Rubin potential scale reduction factor ($\hat{R}$) | | | Effective sample size (ESS) | | |
|---|---|---|---|---|---|---|
| | Average | Min | Max | Average | Min | Max |
| **GRLT (Grain length)** | 0.932 | 0.887 | 0.976 | 1743.15 | 900 | 2593 |
| **GRWD (Grain width)** | 0.982 | 0.947 | 1.018 | 2036.58 | 1143 | 2934 |
| **SDHT (Seedling Height)** | 0.946 | 0.917 | 0.976 | 2009.12 | 1003 | 3023 |

**Table 2. Percentage of non-zero SNP effects (out of 12,486 SNPs) for each model and phenotype, including the frequentist LASSO model.**

| Phenotypes\Model | Spike and Slab | BLASSO | BSLMM | LASSO |
|---|---|---|---|---|
| **GRWD (Grain Width)** | 24.9% | 62.7% | 48.1% | 30.0% |
| **GRLT (Grain Length)** | 38.6% | 69.8% | 59.0% | 28.9% |
| **SDHT (Seedling Height)** | 25.4% | 67.8% | 60.6% | 24.0% |

**Table 3. Evaluation of RMSE, MAE, and Predictive Coverage for Spike-and-Slab, Bayesian LASSO, BSLMM, Ridge, LASSO, and OLS models for GRWD (Grain Width) using a five-fold cross-validation.**

| Methods | RMSE | | | MAE | | | Prediction Coverage | | |
|---|---|---|---|---|---|---|---|---|---|
| | Mean | Min | Max | Mean | Min | Max | Mean | Min | Max |
| **Spike and Slab** | 0.9298 | 0.7437 | 1.1430 | 0.8112 | 0.7078 | 0.8751 | 92.53% | 85.26% | 97.11% |
| **BLASSO** | 1.1230 | 0.9552 | 1.2521 | 0.8810 | 0.7521 | 0.8872 | 82.85% | 76.16% | 84.61% |
| **BSLMM** | 1.0643 | 0.8981 | 1.2863 | 0.9654 | 0.7990 | 0.9073 | 78.15% | 72.29% | 82.48% |
| **Ridge** | 0.8942 | 0.7648 | 1.1168 | 0.8099 | 0.7076 | 0.7926 | – | – | – |
| **LASSO** | 1.0176 | 0.7537 | 1.1763 | 0.8757 | 0.7975 | 0.8553 | – | – | – |
| **OLS** | 1.0383 | – | – | 0.9719 | – | – | – | – | – |

**Table 4. Evaluation of RMSE, MAE, and Predictive Coverage for Spike-and-Slab, Bayesian LASSO, BSLMM, ridge, LASSO and the OLS model for GRLT (Grain Length) using a five-fold cross-validation.**

| Methods | RMSE | | | MAE | | | Prediction Coverage | | |
|---|---|---|---|---|---|---|---|---|---|
| | Mean | Min | Max | Mean | Min | Max | Mean | Min | Max |
| **Spike and Slab** | 0.7361 | 0.5932 | 0.8421 | 0.6895 | 0.5191 | 0.7660 | 92.98% | 86.80% | 97.12% |
| **BLASSO** | 0.9782 | 0.7383 | 1.1138 | 0.7936 | 0.6616 | 0.8961 | 86.64% | 86.19% | 88.30% |
| **BSLMM** | 0.8552 | 0.7399 | 1.0290 | 0.8443 | 0.7463 | 0.8901 | 80.15% | 74.15% | 85.61% |
| **Ridge** | 0.8504 | 0.7663 | 1.1054 | 0.6986 | 0.5805 | 0.7428 | – | – | – |
| **LASSO** | 0.9457 | 0.8712 | 1.0353 | 0.7372 | 0.7177 | 0.8098 | – | – | – |
| **OLS** | 1.0255 | – | – | 0.8993 | – | – | – | – | – |

**Table 5. Evaluation of RMSE, MAE, and Predictive Coverage for Spike-and-Slab, Bayesian LASSO, BSLMM, ridge, LASSO and the OLS model for SDHT (Seedling Height) using a five-fold cross-validation.**

| Methods | RMSE | | | MAE | | | Prediction Coverage | | |
|---|---|---|---|---|---|---|---|---|---|
| | Mean | Min | Max | Mean | Min | Max | Mean | Min | Max |
| **Spike and Slab** | 0.7717 | 0.6335 | 0.9307 | 0.6595 | 0.5322 | 0.7747 | 92.98% | 86.80% | 97.12% |
| **BLASSO** | 0.9212 | 0.8544 | 0.9851 | 0.6981 | 0.5520 | 0.7591 | 86.64% | 86.19% | 88.30% |
| **BSLMM** | 0.8439 | 0.7338 | 0.9767 | 0.6802 | 0.5526 | 0.7959 | 80.15% | 74.15% | 85.61% |
| **Ridge** | 1.0029 | 0.8826 | 1.1138 | 0.6861 | 0.5695 | 0.7750 | – | – | – |
| **LASSO** | 0.8830 | 0.7309 | 1.0136 | 0.6954 | 0.5499 | 0.8085 | – | – | – |
| **OLS** | 0.9954 | – | – | 0.7669 | – | – | – | – | – |

- For Grain Width (Table 3), while the Spike-and-Slab model outperformed BLASSO, BSLMM, LASSO, and OLS, it was surpassed by Ridge regression regarding point estimation accuracy. Ridge regression achieved the lowest Mean RMSE (0.8942) and Mean MAE (0.8099). However, the Spike-and-Slab model retained a decisive advantage in uncertainty quantification, achieving a robust Mean Predictive Coverage of 92.53%, which significantly exceeds the coverage provided by the alternative Bayesian frameworks (BLASSO: 82.85%; BSLMM: 78.15%). Thus, while Ridge offered slightly superior point estimates for this low-variance trait, the Spike-and-Slab model provided more reliable probabilistic inference.

- For Grain Length (Table 4), the Spike-and-Slab model demonstrated better performance, dominating the competing methods across all distributional metrics. The model not only achieved the lowest Mean RMSE (0.7361) and MAE (0.6895), but also demonstrated stability, as indicated by its performance bounds. Notably, the maximum RMSE

recorded for Spike-and-Slab (0.8421) was lower than the mean RMSE of the next best performing method, Ridge regression (0.8504). Furthermore, the model's peak predictive performance was substantial, achieving minimum RMSE and MAE of 0.5932 and 0.5191, respectively, values significantly lower than the respective minima for Ridge (RMSE: 0.7663) and BSLMM (RMSE: 0.7399). This robustness extended to uncertainty quantification, where Spike-and-Slab maintained high coverage rates ranging from 86.80% to 97.12%, providing reliable interval estimates even in the most challenging cross-validation folds. For the OLS estimator, we observed performance trends consistent with the GRWD analysis, which yielded the highest error rates among the compared methods.

- For Seedling Height (Table 5), the Spike-and-Slab model maintained its predictive dominance, outperforming all alternative specifications. It achieved the lowest Mean RMSE (0.7717) and Mean MAE (0.6595), with cross-validation diagnostics highlighting substantial gains in estimation accuracy. Specifically, the model's best-case performance yielded a minimum RMSE of 0.6335 and a minimum MAE of 0.5322, values markedly lower than the respective minima of the closest competitor, BSLMM (Minimum RMSE: 0.7338; Minimum MAE: 0.5526). This superiority extended to interval estimation, where Spike-and-Slab achieved a Mean Predictive Coverage of 92.98%, with fold-specific coverage ranging from 86.80% to 97.12%, significantly exceeding the coverage properties of BLASSO (Mean: 86.64%) and BSLMM (Mean: 80.15%). Regarding the performance of the OLS and Ridge estimators, we observed error trends consistent with the GRWD and GRLT analyses, further reinforcing their limitations in this high-dimensional context.

In Fig 3 we have plotted each of the measures, presenting the various methods and the three phenotypes.

## 5.4 Comparison of the residuals versus prediction plots

In the context of regression diagnostics, residuals versus predicted value plots are crucial in assessing the adequacy of a model. The residuals represent the differences between the observed values and the values predicted by the model. Ideally, in a well-fitting model, residuals should be randomly scattered around zero, with no discernible pattern, indicating that the model captures the underlying data structure well.

In Fig 4, for GRLT, the residuals vs. predicted values for the three models – BLASSO, BSLMM, and Spike and Slab show some subtle differences in performance. The BLASSO model displays a reasonably even distribution of residuals around zero, but slight deviations in the tails suggest some mild heteroscedasticity. The BSLMM and Spike and Slab models appear to show more clustering of residuals near zero, indicating a better fit in capturing the data variance. Notably, BSLMM has tighter residuals, suggesting a potential edge in predictive performance for this phenotype. The residuals scatter widely in the BLASSO case, implying that it may not be the best model for GRLT in this instance.

In the case of GRWD for Fig 5, the residual plots exhibit similar characteristics across all models. The residuals for the BLASSO model display a broader spread, particularly at the extremes, suggesting potential model misspecification or sensitivity to outliers. BSLMM and Spike and Slab exhibit more concentrated residuals around zero, with fewer extreme residuals. The central concentration in the BSLMM model suggests a strong capacity for capturing the core variability in the data. In contrast, the Spike and Slab model shows a similar but slightly broader scatter. For GRWD, these models appear to be more robust than those of BLASSO.

In Fig 6, for SDHT, the residuals are similarly well-concentrated around zero for BSLMM and Spike and Slab, though both models display minor deviations, particularly in the upper ranges. BLASSO, however, presents a slightly wider scatter of residuals, similar to the pattern seen in other phenotypes. The overall tight clustering of residuals around zero in the BSLMM and Spike and Slab models suggests that these models are capable of better capturing the variability in seedling height. In contrast, BLASSO shows more irregularity in the distribution of its residuals, hinting at potential overfitting or inability to model certain parts of the data adequately.

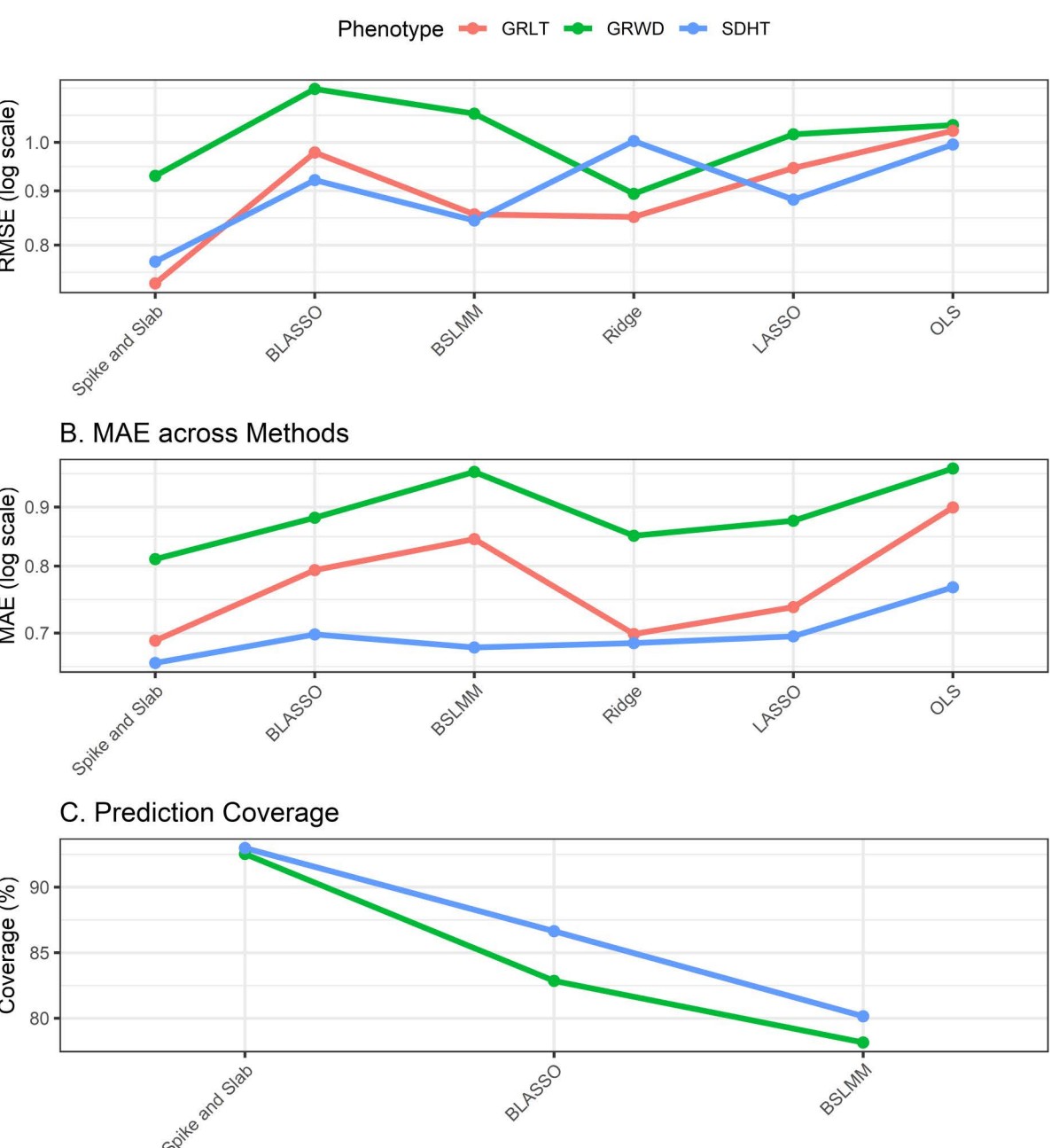

**Fig 3. Comparative assessment of predictive performance.** The figure displays the mean Root Mean Square Error (RMSE; Panel **A**), mean Mean Absolute Error (MAE; Panel **B**), and mean Prediction Coverage (Panel **C**) obtained from five-fold cross-validation. Results are shown for Grain Length (GRLT), Grain Width (GRWD), and Seedling Height (SDHT) across six modeling approaches. Note that Prediction Coverage (Panel C) is reported exclusively for the Bayesian frameworks.

### Residuals vs. Predicted Values for Different Models
Phenotype: Grain Length (GRLT)

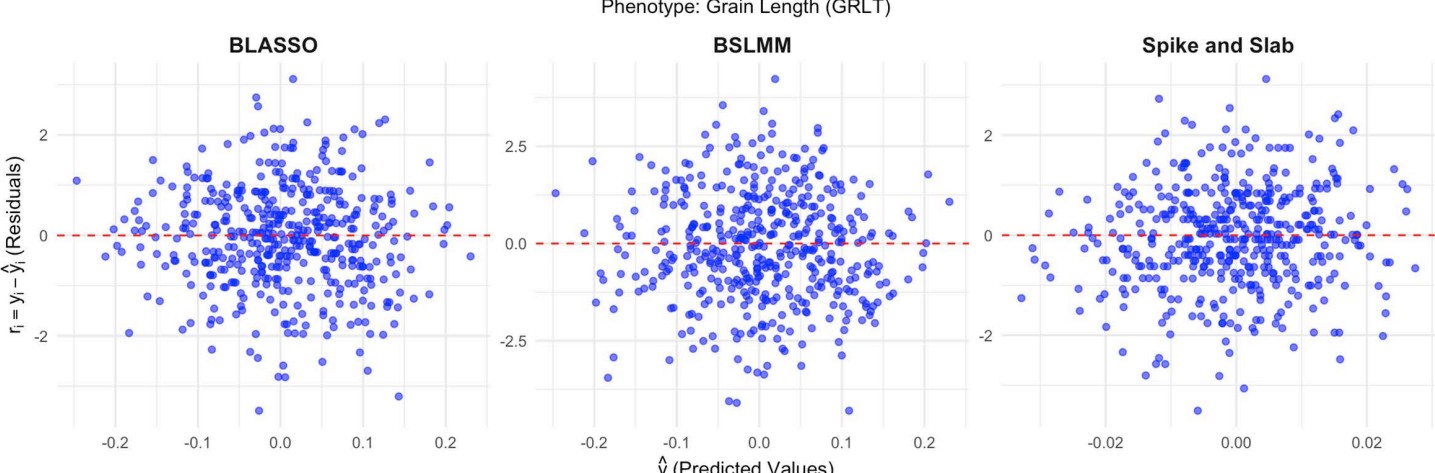

**Fig 4. Residuals vs. Predicted Values for Different Models For Grain Length (GRLT).**

### Residuals vs. Predicted Values for Different Models
Phenotype: Grain Width (GRWD)

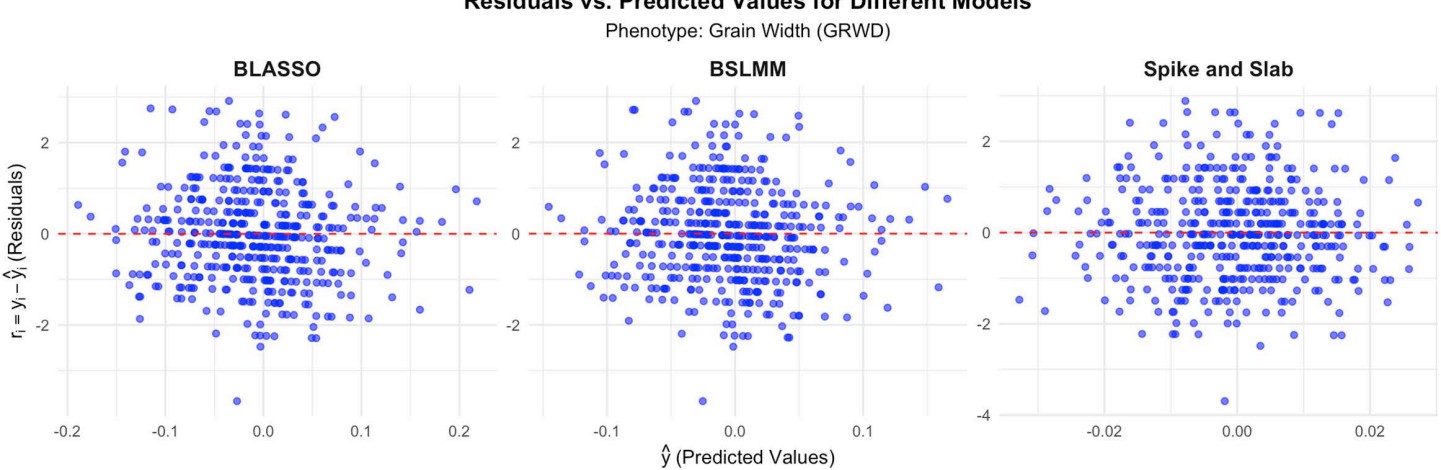

**Fig 5. Residuals vs. Predicted Values for Different Models For Grain Width (GRWD).**

Across all phenotypes (GRLT, GRWD, SDHT), the residuals versus predicted values plots highlight that BLASSO tends to exhibit wider spreads in residuals, indicating it may struggle with capturing the total variance in the data. In contrast, BSLMM and Spike and Slab consistently show tighter, more concentrated residual distributions around zero, implying better predictive performance and robustness for the phenotypes analyzed. These insights suggest that BSLMM and Spike and Slab could be preferable choices in modeling phenotypic traits where precision in predictions is critical.

The results indicate that Spike-and-Slab mostly outperforms other Bayesian models in high-dimensional settings, particularly regarding prediction accuracy, as measured by RMSE and MAE, and predictive coverage. Its ability to dynamically shrink irrelevant variables while retaining the most informative predictors is a key factor in its strong performance. This flexibility is crucial in high-dimensional data scenarios where the number of predictors often exceeds the number of observations, causing overfitting and multicollinearity issues in simpler models like linear regression.

**Residuals vs. Predicted Values for Different Models**

Phenotype: Seedling Height (SDHT)

**Fig 6. Residuals vs. Predicted Values for Different Models For Seedling Height (SDHT).**

BLASSO, while effective in regularizing coefficients, applies uniform shrinkage across all variables, limiting its adaptability in cases where some predictors are significantly more informative than others. This shortcoming may explain its poorer performance relative to Spike-and-Slab, especially in capturing the variability of phenotypes. BSLMM, although incorporating random effects and beneficial for certain traits, struggles to exploit the sparsity in high-dimensional data as effectively as Spike-and-Slab.

The analysis of residuals versus predicted values plots reinforces these findings. Both Spike-and-Slab and BSLMM show tighter residual clustering around zero, indicating their stronger capacity to capture variation in data and improve predictive performance in multiple phenotypes (GRLT, GRWD, SDHT). BLASSO, in contrast, tends to exhibit broader residual distributions, suggesting model misspecification or an inability to effectively capture the underlying data patterns.

## 6 Discussion

In this study, we performed a comparative evaluation of several Bayesian regression models — Spike-and-Slab, Bayesian LASSO (BLASSO), and Bayesian Sparse Linear Mixed Model (BSLMM) — for predicting phenotypic variation. The dataset (as described by [17]) represents a typical high-dimensional genomic setting, with a large number of genotype markers relative to the number of observations, where only a small subset of predictors is expected to contribute meaningfully to phenotypic variation in traits such as grain length, grain width, and seedling height.

The cross-validation results reveal systematic differences in predictive behavior across the competing models. For all error metrics, the results indicate that models explicitly designed to accommodate sparsity and heterogeneity in marker effects tend to provide more stable predictive performance in this setting. In particular, the Spike-and-Slab model demonstrated competitive predictive accuracy across traits, being the best performing method for two out of the three traits, reflecting its ability to adaptively separate informative markers from noise through variable inclusion. BSLMM exhibited intermediate performance, benefiting from its ability to model both sparse effects and a polygenic background, while BLASSO showed comparatively weaker performance in this strongly sparse context.

These findings underscore qualitative differences in how Bayesian models handle shrinkage and sparsity in genomic prediction problems. Methods based on uniform shrinkage, such as BLASSO, may be less flexible when effect sizes are highly heterogeneous, whereas models allowing selective shrinkage can better adapt to datasets where a small number

of markers have relatively stronger effects. The observed differences across models therefore probably reflect structural distinctions in their prior assumptions.

Comparisons with classical regression approaches, including ordinary least squares, LASSO, and ridge regression, further illustrate the limitations of standard methods in GWAS-scale settings with extreme dimensionality. Bayesian formulations that incorporate hierarchical structure and sparsity-aware priors offer a more flexible framework for prediction under such conditions, although their relative advantages depend on the underlying genetic architecture of the trait.

It is important to emphasize that the objective of this work is methodological. Although GWAS-level data are employed, the analysis is framed in terms of genomic prediction and model comparison rather than biological interpretation or locus discovery. As such, no claims are made regarding causal variants or functional relevance of individual markers.

Several avenues for future research emerge from this study. Within the Bayesian framework, alternative prior specifications—such as heavier-tailed distributions or hierarchical priors on inclusion probabilities—may further improve robustness in the presence of rare or large genetic effects. Extending the analysis to additional phenotypes, incorporating interaction effects, and exploring more flexible feature extraction strategies may also enhance predictive performance. Finally, broader comparisons across Bayesian and frequentist methods under consistent cross-validation designs could provide deeper insight into the trade-offs between predictive accuracy, interpretability, and computational complexity in high-dimensional genomic analyses.

## Supporting information

**S1 Text. Basic data exploration including transformations used to normalize the data.**
(PDF)

**S2 Text. Details of posterior distributions and Gibbs Sampling steps.**
(PDF)

**S3 Text. Diagnostics for MCMC convergence.**
(PDF)

## Author contributions

**Conceptualization:** Rupam Basu, Sabyasachi Mukhopadhyay, Kaustubh Adhikari.

**Data curation:** Rupam Basu.

**Formal analysis:** Rupam Basu, Sabyasachi Mukhopadhyay.

**Methodology:** Rupam Basu, Sabyasachi Mukhopadhyay, Kaustubh Adhikari.

**Project administration:** Sabyasachi Mukhopadhyay.

**Supervision:** Sabyasachi Mukhopadhyay.

**Validation:** Rupam Basu.

**Writing – original draft:** Rupam Basu.

**Writing – review & editing:** Rupam Basu, Sabyasachi Mukhopadhyay, Kaustubh Adhikari.

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
