## [Decision Letter · Decision Letter 0]

23 Dec 2025

Dear Dr. Adhikari,

Thank you for submitting your manuscript to PLOS ONE. After careful consideration, we feel that it has merit but does not fully meet PLOS ONE’s publication criteria as it currently stands. Therefore, we invite you to submit a revised version of the manuscript that addresses the points raised during the review process.

We look forward to receiving your revised manuscript.

Kind regards,

Aimin Zhang, Ph.D.

Academic Editor

PLOS One

Journal Requirements:

3. .If the reviewer comments include a recommendation to cite specific previously published works, please review and evaluate these publications to determine whether they are relevant and should be cited. There is no requirement to cite these works unless the editor has indicated otherwise.

Reviewers' comments:

Reviewer's Responses to Questions

**Comments to the Author**

1. Is the manuscript technically sound, and do the data support the conclusions?

Reviewer #1: No

2. Has the statistical analysis been performed appropriately and rigorously?

Reviewer #1: Yes

3. Have the authors made all data underlying the findings in their manuscript fully available?

Reviewer #1: Yes

4. Is the manuscript presented in an intelligible fashion and written in standard English?

Reviewer #1: Yes

Reviewer #1: Recommendation: Major Revision

The manuscript addresses an applied and relevant problem – comparing several Bayesian and frequentist models for genomic prediction in rice using a real, high-dimensional dataset. The overall methodology is standard but appropriate, and the topic fits within the scope of PLOS ONE. However, in its current form, the work falls short of the journal’s standards for methodological transparency and alignment between claims and evidence.

In particular, the main conclusions regarding the superior performance of the spike-and-slab model are somewhat overstated relative to the presented results; the cross-validation design is limited to a single random split; key MCMC/implementation details are insufficiently documented; and the framing in terms of “GWAS” and marker-assisted selection is not fully supported by the analyses.

I therefore recommend major revision, focusing on:

(i) tempering and sharpening the claims to match the actual performance metrics, (ii) strengthening the model comparison design (e.g. repeated or K-fold cross-validation), (iii) providing clearer MCMC diagnostics and implementation details, and (iv) clarifying the scope of the work (genomic prediction/method comparison rather than a full GWAS with biological interpretation).

Minor Issues

1. Section “4.2 Assesing Model Size” should be corrected to “Assessing Model Size.”

2. In Section 3.6, the sentence “measured by the metrics described in Section )” contains an incomplete reference (“Section )”), which appears to be a placeholder that was not updated.

3. The text says “normality of the predictor variables is important…” but the subsequent analysis evaluates phenotype normality, not predictor (SNP) normality. This wording should be corrected for accuracy.

4. Terms such as “Bayesian Sparse Linear Mixed Model (BSLMM)” and “Bayesian sparse linear mixed model” appear with inconsistent capitalization. A unified style is recommended.

5. Section 5.3 repeats conclusions already stated in the RMSE/MAE discussion. Consider condensing the text for clarity.

6. Phrases such as “predicting genotypic variation in Rice” should likely be “predicting phenotypic variation”.

**Do you want your identity to be public for this peer review?** For information about this choice, including consent withdrawal, please see our Privacy Policy

Reviewer #1: No

---

## [Author Response · Author response to Decision Letter 1]

28 Jan 2026

We sincerely thank the Reviewer for their careful reading of our manuscript and for the constructive and insightful comments. We appreciate the time and effort invested in evaluating our work. The suggestions have been extremely helpful in improving the clarity, rigor, and presentation of the manuscript.

In response to the comments, we have undertaken a substantial revision of the manuscript. In particular, we have addressed all the four major points identified by the reviewer: (i) tempered and clarified our claims to ensure they are fully aligned with the reported performance metrics, (ii) strengthened the model comparison framework using a more rigorous validation strategy, (iii) added detailed MCMC convergence diagnostics and implementation details, and (iv) clarified the scope of the study to emphasize its methodological focus on genomic prediction and model comparison rather than biological interpretation of GWAS findings.

All comments from the Reviewer are addressed point by point in detail below. Reviewer comments are reproduced in italic font, followed by our responses. Changes made to the manuscript are indicated with references to the relevant sections and page numbers.

We hope that the revisions adequately address the concerns raised and that the manuscript is now suitable for publication.

Comment 1:

The manuscript addresses an applied and relevant problem – comparing several Bayesian and frequentist models for genomic prediction in rice using a real, high-dimensional dataset. The overall methodology is standard but appropriate, and the topic fits within the scope of PLOS ONE. However, in its current form, the work falls short of the journal’s standards for methodological transparency and alignment between claims and evidence.

In particular, the main conclusions regarding the superior performance of the spike-and-slab model are somewhat overstated relative to the presented results; the cross-validation design is limited to a single random split; key MCMC/implementation details are insufficiently documented; and the framing in terms of “GWAS” and marker-assisted selection is not fully supported by the analyses.

I therefore recommend major revision, focusing on:

(i) tempering and sharpening the claims to match the actual performance metrics,

(ii) strengthening the model comparison design (e.g. repeated or K-fold cross-validation),

(iii) providing clearer MCMC diagnostics and implementation details, and

(iv) clarifying the scope of the work (genomic prediction/method comparison rather than a full GWAS with biological interpretation).

Response: We thank the Reviewer for the careful and constructive evaluation of our manuscript and for recognizing the relevance of the problem and its suitability for PLOS ONE. We have modified our paper according to the changes suggested by the Reviewer.

(i) First, we have revised the Abstract, Results, and Discussion sections to temper and sharpen the claims regarding the performance of the spike-and-slab model. All statements of superiority have been rephrased to accurately reflect the observed performance metrics, emphasizing comparative and context-dependent performance rather than categorical dominance (see revised Abstract, and Results, and Discussion).

(ii) Second, we have strengthened the model comparison framework by replacing the single random train–test split with a more rigorous cross-validation strategy. Specifically, we now employ repeated 5-fold cross-validation to provide a more reliable and stable assessment of predictive performance across models, with all tuning and evaluation conducted within the same validation framework (see Section 4.2 of the modified paper).

“To rigorously assess the predictive performance of the Spike and Slab regression model against alternative Bayesian specifications, as well as frequentist Ridge and LASSO regression benchmarks, we employed a k-fold cross-validation scheme with k = 5. The dataset was randomly partitioned into five disjoint subsets of approximately equal size. In each iteration, four subsets were utilized for model training (representing 80% of the data), while the remaining subset (20%) served as the validation set for out-of-sample prediction. All statistical computations were implemented in the R statistical computing environment. For the Bayesian frameworks, we assessed performance based on RMSE, MAE, and the coverage probability of the 95% posterior predictive intervals. For the frequentist penalized regression models (Ridge and LASSO), performance was evaluated using RMSE and MAE. To provide a comprehensive view of model stability and predictive power, we report the mean, minimum, and maximum values for these metrics across the five folds. The detailed results of this comparative analysis are presented in Section 5.”

The results of the cross-validation exercise and analysis of the same results are discussed in Sections 5.2 and 5.3.

(iii) Third, we have substantially expanded the description of the Bayesian computation and implementation details. This includes explicit reporting of the number of chains, iterations, burn-in, initialization strategy, and sampling scheme, along with a dedicated section on MCMC convergence assessment (see Sections 3.5 - 3.6 in the main text). We have reported standard convergence diagnostics, including trace plots, Gelman–Rubin statistics, and effective sample sizes for key parameters (Section 5.1). Additional diagnostic trace plots and posterior densities of some of the parameters are provided in the Supplementary Material (Supplementary Section S3).

(iv) Finally, we have clarified the scope and framing of the manuscript throughout. While the data originate from a GWAS context, the revised manuscript explicitly positions the contribution as a methodological study focused on genomic prediction and comparative evaluation of high-dimensional regression models, rather than biological interpretation or locus discovery. This clarification has been incorporated in the Introduction and reinforced in a dedicated paragraph in the Discussion on the scope and limitations.

Minor comments:-

1. Section “4.2 Assesing Model Size” should be corrected to “Assessing Model Size.”

We thank the reviewer for pointing this out. We have now corrected this in the modified paper.

2. In Section 3.6, the sentence “measured by the metrics described in Section )” contains an incomplete reference (“Section )”), which appears to be a placeholder that was not updated.

Response: We thank the reviewer for this comment. We have included the Section number in which metrics are described.

We have added the following in current Section 3.7.

“(with deviation measured by the metrics described in Section 4.1)”.

3. The text says “normality of the predictor variables is important…” but the subsequent analysis evaluates phenotype normality, not predictor (SNP) normality. This wording should be corrected for accuracy.

Response: We thank the reviewer for pointing this out. We have now corrected this sentence as “Since normality of the regressed variables is an important…”.

4. Terms such as “Bayesian Sparse Linear Mixed Model (BSLMM)” and “Bayesian sparse linear mixed model” appear with inconsistent capitalization. A unified style is recommended.

Response: We thank the reviewer for pointing this out. We have now followed a unified style as Bayesian Sparse Linear Mixed Model.

5. Section 5.3 repeats conclusions already stated in the RMSE/MAE discussion. Consider condensing the text for clarity.

Response: We have condensed the Discussion to avoid repetition of the detailed RMSE/MAE analysis presented in Section 5.3. The revised Discussion now summarizes cross-validation findings at a higher level, emphasizing comparative predictive behavior and methodological insights rather than metric-specific details.

6. Phrases such as “predicting genotypic variation in Rice” should likely be “predicting phenotypic variation”.

Response: We thank the reviewer for this suggestion. We have made the change accordingly in the modified manuscript.

---

## [Decision Letter · Decision Letter 1]

15 Feb 2026

Bayesian Variable Selection for Genome-Wide Association Study of Grain Traits in Rice

PONE-D-25-56999R1

Dear Dr. Adhikari,

We’re pleased to inform you that your manuscript has been judged scientifically suitable for publication and will be formally accepted for publication once it meets all outstanding technical requirements.

Kind regards,

Aimin Zhang, Ph.D.

Academic Editor

PLOS One

Additional Editor Comments (optional):

Reviewers' comments:

Reviewer's Responses to Questions

**Comments to the Author**

Reviewer #1: All comments have been addressed

2. Is the manuscript technically sound, and do the data support the conclusions?

Reviewer #1: Yes

3. Has the statistical analysis been performed appropriately and rigorously?

Reviewer #1: Yes

4. Have the authors made all data underlying the findings in their manuscript fully available?

Reviewer #1: Yes

5. Is the manuscript presented in an intelligible fashion and written in standard English?

Reviewer #1: Yes

Reviewer #1: (No Response)

**Do you want your identity to be public for this peer review?** For information about this choice, including consent withdrawal, please see our Privacy Policy

Reviewer #1: No

---

## [Editor Report · Acceptance letter]

PONE-D-25-56999R1

PLOS One

Dear Dr. Adhikari,

I'm pleased to inform you that your manuscript has been deemed suitable for publication in PLOS One. Congratulations! Your manuscript is now being handed over to our production team.

Kind regards,

on behalf of

Prof. Aimin Zhang

Academic Editor

PLOS One